# Progress in the Preparation, Properties, and Applications of PLA and Its Composite Microporous Materials by Supercritical CO_2_: A Review from 2020 to 2022

**DOI:** 10.3390/polym14204320

**Published:** 2022-10-14

**Authors:** Kangming Peng, Suhail Mubarak, Xuefeng Diao, Zewei Cai, Chen Zhang, Jianlei Wang, Lixin Wu

**Affiliations:** 1CAS Haixi Industrial Technology Innovation Center in Beilun, Ningbo 315830, China; 2College of Chemistry and Materials Science, Fujian Normal University, Fuzhou 350007, China; 3CAS Key Laboratory of Design and Assembly of Functional Nanostructures, Fujian Key Laboratory of Nanomaterials, Fujian Institute of Research on the Structure of Matter, Chinese Academy of Sciences, Fuzhou 350002, China; 4Department of Chemical and Biomolecular Engineering, Chonnam National University, Yeosu-si 59626, Jeonnam, Korea; 5Jinyoung (Xiamen) Advanced Materials Technology Co., Ltd., Xiamen 361028, China; 6College of Chemistry, Fuzhou University, Fuzhou 350116, China; 7School of Materials and Chemistry Engineering, Minjiang University, Xiyuangong Road No. 200, Fuzhou 350108, China; 8Industrial Design Institute, Minjiang University, Xiyuangong Road No. 200, Fuzhou 350108, China

**Keywords:** foaming, poly(lactic acid), carbon dioxide, modification, review

## Abstract

The development of degradable plastic foams is in line with the current development concept of being pollution free and sustainable. Poly(lactic acid) (PLA) microporous foam with biodegradability, good heat resistance, biocompatibility, and mechanical properties can be successfully applied in cushioning packaging, heat insulation, noise reduction, filtration and adsorption, tissue engineering, and other fields. This paper summarizes and critically evaluates the latest research on preparing PLA microporous materials by supercritical carbon dioxide (scCO_2_) physical foaming since 2020. This paper first introduces the scCO_2_ foaming technologies for PLA and its composite foams, discusses the CO_2_-assisted foaming processes, and analyzes the effects of process parameters on PLA foaming. After that, the paper reviews the effects of modification methods such as chemical modification, filler filling, and mixing on the rheological and crystallization behaviors of PLA and provides an in-depth analysis of the mechanism of PLA foaming behavior to provide theoretical guidance for future research on PLA foaming. Lastly, the development and applications of PLA microporous materials based on scCO_2_ foaming technologies are prospected.

## 1. Introduction

At present, the polymer foam products widely used in production and life are mainly composed of petroleum-based plastics, such as polyethylene (PE) [1], polystyrene (PS) [2], and poly(methyl methacrylate) (PMMA) [3]. The extensive use of petroleum-based foam, on the one hand, leads to the excessive consumption of petroleum resources, making environmental problems more serious. Bio-based plastics offer a possible solution to preventing these problems [4]. PLA is a typical biodegradable plastic extracted from biological crops (such as starch and corn stover) [5,6]. The ester group on the molecular chain of PLA makes it have good degradability. It has been reported that under controlled composting conditions, PLA can be broken down into water and CO_2_ in less than 90 days, while microorganisms can completely assimilate the degradation products [4,7]. PLA contains a certain proportion of L-lactate monomer and D-lactate monomer [8,9]. When the D-lactic acid in PLA exceeds 12%, it no longer crystallizes and becomes amorphous. Compared with PLA with high D-lactic acid content, PLA with low D-lactic acid content is more conducive to forming a good pore structure at higher foaming temperatures due to its high crystallinity under the same other foaming conditions. [10,11,12,13]. It is one of the most widely used biodegradable polymers. However, the application and development of PLA-based foams are greatly limited due to the low melt strength, slow speed of crystallization, and high brittleness of PLA [14,15,16]. The strength of the melt is too low, and the entanglement degree is low in the melting state. In the process of foaming, the bubbles easily merge, form large bubble structures, or even break, thus reducing the material’s mechanical properties. The melt strength is too high to obtain foam with a high expansion ratio. Therefore, building the right melt strength through the process is the main research problem related to PLA foaming. On the other hand, proper crystallization plays the role of heterogeneous nucleation, which can improve the strength of the melt, which is conducive to the formation of high cell density. However, the high crystallinity is unfavorable to the dissolution of the foaming agent, and the crystal zone cannot dissolve the foaming agent, resulting in the gas being more inclined to enter the bubble rather than nucleation, which is not conducive to foaming. Therefore, the PLA characteristics of low melt strength and slow crystallization are important reasons for limiting the large-scale production of polylactic acid foam microporous foam. The main idea to improve the foaming behavior and foaming ability of PLA is to adjust the rheological properties and microstructure of the polymer by modification. The modification methods include chemical modification [17,18], filler modification [19,20], blending [21], and other modification methods [22].

Supercritical fluid foaming technology using CO_2_ or N_2_ as a physical foaming agent is a green and efficient polymer foam processing technology, as well as a microporous foaming technology [23,24,25,26,27]. Microporous foams are defined as porous materials with 10 μm or even smaller dimensions [28], and the tiny pores give the material good impact resistance and adsorption properties. Currently, more studies on PLA foam use supercritical CO_2_ as a physical blowing agent. The basic principle of this process is as follows: The gas diffuses into the polymer to form a polymer/gas two-phase system, and when the gas reaches saturation, the system is homogeneous. Then, a sudden rise in temperature or a sudden drop in pressure leads to the supersaturation of the gas in the system, which then triggers the nucleation of the cells. Cell growth occurs when the gas diffuses into the nucleus. Finally, the bubble structure is stabilized by cooling to obtain foam. The diffusion and dissolution behavior of CO_2_ in polymers is complex, and the intrinsic mechanism of the polymer and CO_2_ has not been systematically explained. Firstly, CO_2_ has a plasticizing effect on the polymer, which can reduce the glass transition temperature (T_g_) and melting temperature (T_m_) of the polymer and affect the crystallization behavior of the polymer [29,30,31,32]. Secondly, in the nucleation stage, the supersaturated promotes nucleation at the crystal/amorphous interface of the polymer system [33,34]. Due to its light weight, material saving property, heat and sound insulation, good impact resistance, and other advantages, PLA foam has broad application prospects in the fields of heat preservation and insulation, noise reduction and sound insulation, oil absorption and adsorption [35,36], electrostatic packaging, biomedical [37] and other fields [38,39,40].

Figure 1 shows the PLA foam research activity curve for the decade 2012 to 2022. The graph was generated from Web of Science results, with the keywords “foam”, “CO_2_”, and “PLA”. Since PLA foaming research has been reviewed in the literature before 2020 [41,42,43], this paper focuses on the progress of the preparation of PLA microporous materials based on scCO_2_ foaming technology since 2020. After a careful literature search and keyword qualification, we identified 71 articles that fall within the scope of this review using Web of Science.

In the last three years, the main activity in the foaming of PLA with CO_2_ as a physical foaming agent continues to be the investigation of the effect of modification on PLA-based foaming capacity and properties. In the last three years, the common theme surrounding the preparation of PLA microcellular foams with supercritical CO_2_ has been the effect of the choice of nucleating agent on the foaming behavior. Most of the foaming experiments used pressure-induced or temperature-induced intermittent foaming. A few articles also report on continuous foaming experiments, such as microporous injection foaming, extrusion foaming and 3D printing foaming. This paper reviews the modification of PLA composite foams and the scCO_2_ foaming processes in the past three years. An outlook on future applications of PLA foaming is provided. In the following sections, we will review the progress of PLA microcellular foams in three aspects: scCO_2_ foaming technologies, the effect of modification on the foaming of PLA and its composites, and the applications of PLA composite foams.

## 2. Supercritical CO_2_ Foaming Technologies

PLA foaming is divided into physical foaming and chemical foaming [44,45,46]; in contrast with the latter, the use of a physical foaming agent for PLA foaming has the characteristic of being green and non-polluting in line with the current carbon-neutral development plan. At the same time, the foam obtained by physical foaming has the properties of lightweight, low density, and more stable. Common physical blowing agents are CO_2_ and N_2_. Due to the plasticizing effect of CO_2_, and its high solubility in PLA, which can promote the crystallization of PLA, the current research on the supercritical foaming of PLA, especially intermittent foaming, mainly uses CO_2_ as the preferred foaming agent. However, due to the fast diffusion rate of N_2_, smaller bubbles can be obtained in microcellular injection foaming using N_2_. Therefore, N_2_ is commonly used as a blowing agent in the microcellular injection foaming process [47]. In the supercritical foaming process, foaming parameters, such as saturation temperature, saturation pressure, and saturation time have a great influence on the structure and properties of the bubble pores. The cell diameter, cell density, and foam volume expansion ratio are three fundamental parameters for characterizing the cell structure. The variation in the three parameters has a great influence on the cell structure and the performance of the foam [48]. Cell diameter generally refers to the average diameter of at least 100 cell units in the foaming image obtained from electron microscopy [49]. Cell density refers to the number of cells per cubic centimeter of the foamed sample. Volume expansion ratio refers to the density ratio of the unfoamed sample to the foamed sample.

The current technologies for the preparation of PLA foam by supercritical foaming mainly include batch foaming technology, extrusion foaming technology, and microporous injection molding foaming technology, in addition to new foaming technologies such as 3D printing foaming [50].

### 2.1. Batch Foaming Technology

Batch foaming involves saturating the polymer with a supercritical fluid in an autoclave for a while and then inducing polymer cell nucleation with rapid pressure reduction or rapid temperature increase, resulting in cell growth by gas diffusion. A sudden drop in pressure or a sudden temperature rise causes the system to change from a homogeneous system to a thermodynamically unstable state, thus inducing cell nucleation. Before saturation is generally low-pressure purging of the reactor, after foaming to ensure the stability of the bubble pore structure requires the operation of cooling the foam. Batch foaming is mainly divided into the pressure-induced foaming method and temperature-induced foaming method. Table 1 summarizes the research on the preparation of PLA and its composite foams using the batch foaming method in the past three years. Most of the PLA foams were prepared by batch foaming by pressure induction, and a few researchers used temperature-induced foaming. The following review addresses the studies in the table.

Pressure-induced foaming refers to the foaming method in which the polymer is saturated in a reaction kettle at a certain temperature and pressure for a certain time and then the pressure is quickly released to form microporous directly. The foaming process is shown in Figure 2.

Wang et al. [72] prepared polycaprolactone (PCL)/PLA composites using pressure-induced foaming at different foaming temperatures (foaming temperatures: 45 °C, 50 °C, 55 °C, and 60 °C). As the foaming temperature increased, the pore diameter increased, the pore density decreased, and the pore wall became thinner or even ruptured. The increase in temperature decreased the PLA melt strength and gas solubility, which were insufficient to support the growth of cells. Then the pores merged or even collapsed, increasing the cell opening rate. Similar conclusions were obtained by Li et al. [56]. They used a pressure-induced method to prepare PLA/poly(butylene succinate) (PBS) composite foam. The cell opening rate decreased, and the cell diameter decreased with the decrease in foaming temperature.

It is noteworthy that the pressure-induced foaming method is the primary foaming means of PLA foams studied so far in the last three years. Some researchers have optimized the one-step method mentioned above and implemented a two-step foaming method. Li et al. [57] prepared PLA foams by combining pre-melting and pressure-induced foaming. They investigated the effects of different foaming temperatures and pressures on the pore morphology and material properties. The pre-melt treatment can ensure the CO_2_ adsorption equilibrium. The results show that closed or open cell foam can be obtained under different foaming processes by optimizing the intermittent foaming process, which provides a theoretical basis for applying functional PLA foams. The effect of foaming temperature on the morphology of the foam pores was similar to the results of the above article [56,72]. As the saturation pressure increases, the cell size gradually decreases, and the cell density increases. This result is thought to be caused by the increased solubility of CO_2_ in PLA. In this article, the authors also pointed out that the saturation pressure of the foaming process has a more significant effect on the morphology of the pores than the foaming temperature.

Interestingly, for unmodified PLA, spherical crystals appear at low-temperature foaming and disappear as the foaming temperature increases, foaming in an amorphous state. Another paper from the same research group [57] reported the preparation of unmodified PLA foams by pre-crystallization combined with pressure-induced foaming. In the pre-crystallization process, PLA can form lamellar structures at low temperatures and transform into non-lamellar structures at higher temperatures. Therefore, different foam morphology can be obtained at different foaming temperatures. As the saturation pressure increases, the migration rate of amorphous molecular chains increases and crystallization decreases, leading to interfacial surface tension and melt strength being insufficient to support the growth of cells, and cell walls become thinner. Cells merge to form high open-cell rate foams.

As shown in Figure 3, temperature-induced foaming means that the polymer is first saturated in the reactor at low temperature and low pressure and then removed from the reactor and quickly put into a certain temperature medium (usually hot water, oil, etc.) for heat bath-induced foaming. When choosing water bath-induced foaming, the ease of hydrolysis of PLA should be taken into account, which may impact the performance of the foamed material, although temperature-induced foaming is a relatively short process. Sun et al. [67] prepared PLLA/PBS/PDLA composite foam by temperature-induced intermittent foaming. The effects of the introduction of poly(butylene succinate) (PBS) and PDLA on the crystallization behavior and cellular structure of PLLA were investigated. It was found that the introduction of both polymers resulted in the formation of stereo-complex crystals in the PLLA matrix. This stereo-complex structure, while limiting the movement of molecular chains, reduced the nucleation rate of PBS. However, it improves the compatibility of PLLA/PBS composites, plays a positive role in foaming, refines the micropore size, and provides the idea of using PLA stereo-crystals to improve the foaming behavior of PLA. Zhang et al. [71] prepared CA/PLA composite foams using a temperature-induced intermittent foaming method to investigate cellulose acetate foam’s properties (CA). The results illustrated that the addition of PLA reduced the system’s storage modulus and complex viscosity, increased the loss factor, and improved the rheological properties of CA. The cell size increased, and the impact strength of the foam slightly decreased with the increase of PLA content.

### 2.2. Extrusion Foaming Technology

Figure 4 shows the principle of polymer extrusion foaming. Extrusion foaming enters polymer particles through the feed port and melting plasticization through the screw. When passing through the homogenizing section of the screw with a physical blowing agent (supercritical CO_2_ on the figure), molten PLA and supercritical fluid form a homogeneous system, and the extrusion process has a sudden pressure drop, which puts the homogeneous system in a thermodynamically unstable state and induces bubbles to generate. After extrusion, the cells grow. Finally, a cooling device is provided to stabilize the cell structure. For the better dissolution of the gas in the polymer, an oil bath can be used for a static mixing process prior to extrusion. Since extrusion foaming is a continuous process, it is currently one of the primary methods for the industrial production of PLA foams, the other being microporous injection molding.

In the extrusion foaming process, extrusion process parameters such as processing temperature, extrusion outlet die pressure and screw speed are essential factors affecting the structure and properties of the bubbles. Chauvet et al. [74] obtained PLA/thermoplastic starch (TPS) composite foams using the extrusion foaming process and investigated the effect of die temperature on the bubble morphology of the composite foams. The results showed that the composite foams with 20 wt% TPS content showed a decrease in nucleation rate, a decrease in cell size, and an uneven distribution of bubbles as the die temperature increased. The increase in die temperature decreases the melt strength of the system and generates gas escape, which can cause the cells to merge and collapse in severe cases. In addition, due to the amorphous nature of thermoplastic starch, the lower glass transition temperature is also conducive to gas escape. The uneven distribution of the TPS structure at high temperatures can cause uneven distribution of the final vesicles. The authors also confirmed the poor compatibility between pure thermoplastic starch and PLA/TPS composites containing 50 wt% TPS by studying the bubble morphology.

Sadeghi et al. [73] added hydrophilic cellulose nanofibers (CNF) to PLA and prepared microporous foams by twin-screw extrusion foaming and found that the addition of CNF promoted cell nucleation and increased cell density. The molecular weight of pure PLA was reduced by about 25% after extrusion, probably due to the hydrolysis of ester groups that occurred during the melting process. The higher the extrusion outlet die pressure, the more significant the pressure difference in the extrusion process and the higher the pressure plunge, which favors the nucleation rate. It has been shown that higher levels of CO_2_ promote foaming during the extrusion process [75].

### 2.3. Microporous Injection Molding Foaming Technology

Microporous injection molding (MIM) foam technology was first patented by Trexel Corporation in 1997 and applied to industrial production. The molding principle is similar to the polymer injection molding process. The difference is that a supercritical fluid module is added to the MIM injection process that is mixed by a screw to form a molten polymer/supercritical fluid homogeneous system that is then injected into the mold. Depending on the mold, it can be divided into low-pressure and high-pressure injection molding. When the gas/polymer melt enters the mold, it immediately undergoes a pressure drop. Foaming occurs instantaneously and expands in the mold cavity, called low-pressure injection molding (a short shot). The mold is filled with the gas/polymer melt at high pressure, and during the mold opening process, the rapid pressure relief induces bubble formation, followed by the growth of the gas-filled cell, which is called high-pressure injection molding (full shot). The foam obtained by injection foaming has sandwich structure characteristics, as shown in Figure 5; its microscopic bubble structure consists of a harmful foaming effect epidermal layer and a foaming core layer composed of different bubbles [76]. In the process of injection molding, the pressure, holding time [77,78], opening speed and mold opening distance (for high pressure injection foam) have great influence on the foam structure after molding.

Li et al. [80] used chemical injection foaming technique to prepare PLA/random terpolymer (ethylene, acrylate, and glycidyl methacrylate) composite foams with improved foam morphology. Wang et al. [79] explored a new in situ fiber reinforcement combined with high-pressure microporous injection molding (HPMIM) technique for the preparation of PLA composite foams. In this work, in situ nanofibrillation plays the role of chain expansion/cross-linking, and nanofiber poly(ethylene terephthalate) (PET) with 10 wt% content acts as reinforcement and CO_2_ as blowing agent to prepare a composite foam with high impact strength and tensile strength and good thermal insulation. The differences between the morphology and properties of foams obtained by HPMIM and regular microporous injection molding (RMIM) methods were compared under the same injection foaming parameters (melt temperature, mold temperature, injection speed, gas dosage), and the results showed that the improved cell morphology made by HPMIM was attributed to the decoupling of the melt filling process and the enhanced melt strength under high pressure conditions. The high-pressure holding time affects the cell morphology and the PLA/PET composite foam cell morphology prepared by HPMIM is significantly improved compared to that of pure PLA, which is shown to be the result of the action of nanofiber PET. It is of interest that the PLA/PET foam made by this newly developed process has an expansion multiplicity (26.2) that is the maximum obtained by foam injection molding at that time.

In a similar study, Zhao et al. [81] prepared PLA/ethylene-propylene terpolymer (EPDM) composites using in situ nanofibrillation combined with ultraviolet (UV) crosslinking. The experimental procedure was to construct a PLA/EPDM fibrillar structure by melt blow molding and curing the EPDM fibrillar structure using UV crosslinking. The PLA/PEPDM composite foam was obtained using a high-pressure injection molding technique. UV crosslinking acted as a curing agent to further enhance PLA crystallization and melt strength. The authors also investigated the effect of mold opening distance on the bubble morphology, and the mold opening distance had a small effect on the expansion multiplicity of pure PLA foam. As the opening distance increased (at 5 mm to 35 mm), the cell stretching along the opening direction increased, affecting the cell growth orientation. Like the results of the previous study, microcellular injection foaming combined with in-situ fibrillation resulted in high expansion multiplicity PLA/EPDM composite foams (maximum expansion multiplicity up to 28 times). Meanwhile, in situ fibrillation can obtain microporous materials with high toughness and high impact strength, which has great potential for commercial application.

### 2.4. Other Foaming Technologies

3D printing foaming has been introduced as a foaming technology for the future [82]. 3D printing’s high degree of freedom in design enables the design of complex and diverse porous structures to meet the characteristics of lightweight porous materials [83]. PLA 3D printing foaming technology [84,85,86], which generally uses fused deposition printing (FDM) foaming, is based on the principle of introducing supercritical fluid at the nozzle of FDM to induce foaming during printing. However, this process is still in its embryonic stage. In the FDM printing process, a systematic solution has not yet been established for how to introduce gas into the nozzle in a stable manner or how to saturate the gas in a small diameter wire and not to escape as well as for the bonding between layers during the printing process. The current application of 3D printing technology for foaming generally means that the sample is printed and processed first and then foamed using intermittent foaming methods, which is only applicable to scientific research and challenging to produce [87,88].

### 2.5. Conclusions

This paper reviews the technology for preparing PLA and composite foams under supercritical CO_2_. The following conclusions can be obtained:

For supercritical CO_2_ foaming technology, within the foaming window and lower foaming temperature, PLA facilitates the formation of smaller size, high-density cells, whose cell structure tends to be closed pore type. Higher foaming temperature, melt strength decreases, gas escapes, vesicles merge or even collapse, more extensive cell size, vesicle density decreases, and open-pore rate increases. At the same time, higher saturation pressure, which helps the dissolution of CO_2_ adsorption, is conducive to nucleation. During the foaming process, the pressure relief or temperature increase rate is significant, which is also favorable to cell nucleation. The modification of PLA also affects supercritical CO_2_ foaming, and its influence mainly affects the bubble morphology and performance by regulating the melt strength and crystallization behavior, which will be reviewed in the next section.

For batch foaming, the degree of crystallization affects the contribution of melt strength to the foaming process. For unmodified PLA foams, lower foaming temperatures have spherical crystal generation. Near the spherical crystals, the bubbles tend to nucleate and grow between the crystals. The temperature increases further, and the crystallinity decreases, both of which affect the bubble morphology. Currently, the preparation of PLA foam based on supercritical CO_2_ assisted is mainly obtained by batch foaming, which can prepare foams with higher expansion multiplicity and complex structure. At the same time, the foaming process is adjustable. For example, two-step foaming is an improvement of one-step foaming, and the addition of processes such as pre-crystallization or pre-melting can play a significant role in the crystallization behavior and melt strength of the PLA foaming process, optimizing the foam morphology and properties. However, it has a long production cycle and discontinuous foaming, making it challenging to promote the technology on an industrial scale.

The most critical process parameter affecting extrusion foaming is die temperature. Extrusion die opening pressure and screw speed can also have an effect. According to the results of related literature, different PLA blend systems can be analyzed to obtain a process that is favorable to the foam morphology and properties. Like extrusion foaming, microcellular injection foaming is continuous foaming and has a vast prospect for commercial applications. For preparing PLA and its composite foams by microcellular injection foaming, the parameter regulation of the mold opening process and the choice of pressure are the main factors affecting the bubble morphology and properties. Meanwhile, developing a more optimized preparation process combined with microporous injection foaming is an important research direction to promote PLA foam commercialization. With further research on the bonding between 3D printed foam layers, it is of great significance to prepare more lightweight foams.

## 3. Effect of Modifications on Foaming of PLA Composites

The low melt strength and poor crystallization ability are the reasons for the narrow processing window of PLA foaming and therefore essentially limit the foaming ability of PLA. Currently, most researchers are still focusing on modifying PLA to regulate its melt strength and crystallization ability and thus improve its foam morphology and properties. The modifications include chemical modification, filler modification, fiber reinforcement, and blending.

### 3.1. Chemical Modification

Table 2 summarizes the research on chemical modification of poly(lactic acid)-based foams in the last three years. Chemical modification refers to a modification by changing the type of atoms or groups of atoms on the polymer macromolecule chain and the way of bonding through chemical reactions. For PLA foams, chemical modification is mainly carried out by chain expansion or cross-linking. PLA chain expansion refers to the reaction of hydroxyl or carboxyl groups on PLA chains with functional molecules containing hydroxyl, carboxylic acid, anhydride, isocyanate, and other functional groups to produce branching. The introduction of cross-linking agents can transform linear or lightly branched macromolecules into a three-dimensional network structure and improve material properties. The generation of branched chains in PLA by adding chain extenders can effectively improve the melt strength of PLA. At the same time, branching plays a role similar to heterogeneous nucleation and affects the crystallization behavior of PLA. However, through chemical modification, the biodegradability and recyclability of PLA can be affected. For example, the cross-linking process produces gels [89].

Ni et al. [54] prepared poly(lactic acid)-based composites using multifunctional epoxy-based styrene-acrylic acid oligomers with 5 wt% content as chain extenders (CE). The effect of crystals generated by the cold crystallization process on the cell size of PLA foam was investigated. The authors used the method of quenching after blending to obtain thin sheets to avoid the generation of crystals before foaming, while intermittent foaming was carried out at 15 MPa and saturation temperature from 85 °C to 120 °C. The results showed that the transition from nano to micron foam cell structure could be induced at the cold crystallization temperature. Compared with pure linear PLA, the melt strength was enhanced, the volume expansion ratio (VER) was significantly increased, and the micro-nano transition temperature was reduced from 117 °C to 115 °C after chain expansion. The micro/nano transition is due to the cold crystallization process, where the temperature induces a crystal transition from α′ to α. Imperfect α′ crystals produced below the micro/nano transition temperature provide more efficient nucleation sites and promote nano-foam generation.

Similar conclusions were reached by Zhang et al. [92] in their study of the thermal behavior of poly(l-lactide) (PLLA). PLLA undergoes a disordered-ordered (α′-α) phase transition near 120 °C, and the crystal transition affects the enthalpy change of the melt. In a similar study, researchers [90] added different levels (0, 2 wt%, 4 wt%, 6 wt%, 8 wt%) of multifunctional epoxy-based CE (random copolymer of ethylene and glycidyl methacrylate) as chain extenders to PLLA to obtain PLLA foams with high volume expansion. After chain expansion, PLA produces branching and cross-linked structures, which improve melt viscosity and crystallization behavior. At higher foaming temperatures (115 °C), the VER is close to 35 times.

There are three main reasons for chain expansion as one effective modification means during PLA foaming: Firstly, after chain expansion, the PLA macromolecular chain produces branches to form branching PLA. Due to the generation of a branching structure, the relative molecular weight of PLA increases, and the molecular structure changes, dramatically impacting PLA’s foaming behavior [93]. Secondly, chain expansion can change the rheological behavior of PLA in the molten state and increase the melt storage modulus, which can effectively improve the melt strength of PLA and then enhance the foaming effect. Finally, in the PLA foaming process, the branched or cross-linked structure acts as a heterogeneous nucleation, which makes PLA more crystalline, improves the crystallization behavior and enhances the performance of the foam.

In [91], the authors used acrylic acid oligomer as a chain extender and self-assembled dibenzoylhydrazide decanoate dibenzoylhydrazide (TMC-300) as a crystalline nucleating agent to prepare PLA foams with different contents of TMC-300. The experiment was performed using batch foaming. It was observed that the addition of TMC-300 significantly improved the crystallization rate and melted strength of the branched PLA compared with the branched PLA after the addition of the chain extender, the microporous foam density was able to increase by two orders of magnitude, and the cell size was reduced to 10 µm which effectively improved the foaming ability of PLA. This was attributed to the addition of TMC-300, which forms local stress and increases the number of nucleation sites. The fibrous skeleton formed at the same time further promotes crystallization (Figure 6). The discontinuity at the interface between the two phases and the cell growth process receive biaxial stretching of the cell wall, which facilitates obtaining foams with a high expansion ratio.

In order to effectively improve the foaming ability of PLA, Zhao et al. [81] added crosslinker (1,3,5-Triallyl-1,3,5-triazine-2,4,6 (1H,3H,5H)-trione) and photoinitiator to PLA and ethylene propylene terpolymer (EPDM) blends to prepare PLA/EPDM multifunctional ultra-high expansion rate nanocomposite foam materials, which broadens the industrial applications of PLA foam materials. The authors developed a new method combining in situ nanofibrillation and UV crosslinking to obtain in situ nanofibers, which overcame the problem that introducing traditional fiber fillers was difficult to effectively enhance the material’s toughness. Injection molding was used to obtain foams with a high expansion ratio (28 times), high toughness (620% improvement in toughness compared with pure PLA), and good thermal insulation properties.

Functionalized PLA foam is also a research direction. Jia et al. [60] prepared ultra-low thermal conductivity PLA/SO_2_ aerogel composite foam at saturation temperature of 130 °C, saturation pressure of 20 MPa, and saturation time of 4 h. The authors used diphenylmethane diisocyanate (MDI) for PLA chain expansion, which could improve the rheological behavior of PLA to a greater extent. The results showed that adding MDI increases cell nucleation rate. The introduction of aerogel further improved the melt viscosity of PLA under melting. The cell size decreased, the density of cell increased, the thermal conductivity decreased significantly, and the ultra-low-density foam with good thermal insulation was successfully prepared.

### 3.2. Filler Modification

Table 3 shows the research on filler modified polylactic acid foam from the last three years. Adding fillers, mainly nano-scale fillers, to PLA is an effective way to improve PLA foams:

(1)Filler modification such as with natural fibers and talcum powder is simple and easy in terms of preparation.(2)Nano-scale fillers are an ideal heterogeneous nucleating agent that can greatly enhance the crystallization ability of PLA and improve cell density.(3)In terms of melt viscoelasticity, fillers can usually play a role in enhancing the strength of the melt and improving the cell structure and mechanical properties.

**Table 3 polymers-14-04320-t003:** Study on modification of PLA foam by fillers in recent three years.

Research System	PLA Brand	Expansion Ratio	Cell Size	Cell Density	Reference
PLA/Cotton fiber	2003D	--	65 µm	3 × 10^8^ cell/cm^3^	[94]
PLA/MCF	4060D	~5	24 µm	~3 × 10^7^ cell/cm^3^	[52]
CA/PLA	4032D	--	0.6 µm	~10^12^ cell/cm^3^	[71]
PLA/CNF	8052D	~3	~100 µm	--	[73]
PLA/CNF	TP-4000	20.4	<40 µm	~10^9^ cell/cm^3^	[70]
PLA/CNC	4060D	--	~50 µm	5 × 10^6^ cell/cm^3^	[69]
PLA/CNT/CE	2003D	~49.6	~90 µm	~10^7^ cell/cm^3^	[95]
PLA/CNTs/CB	4032D	~3	~13 µm	10^8^ cell/cm^3^	[62]
PLA/HNA	3052D	~6	~6.59 µm	~10^9^ cell/cm^3^	[55]
PLA/TPS	PLE001	~5	~250 µm	--	[74]
PLA/HNTs	4032D	~19	~150 µm	--	[61]
PLA/SiO_2_ aerogel/CE	4043D	~55	120 µm	0.024 g/cm^3^	[60]

In the table, “CE” refers to chain extender.

Sadeghi et al. [73] reported preparing PLA/CNF composite foam materials by adding an appropriate amount of cellulose nanofibers (CNF) to PLA. The continuous extrusion foaming method was used to avoid the shortcomings of the discontinuous production of intermittent foaming and to meet the requirements of industrialization. Intriguingly, the authors added a static mixing process before foaming to ensure the particle dispersion was as uniform as possible to better mix the nanofibers with the matrix. The addition of CNF greatly enhanced the loss factor and improved the PLA rheological properties. The foam obtained with the addition of CNF with a content of 1.5 wt% and foaming pressure maintained at 30 bar exhibited a high expansion rate (3.149). It indicates that CNF is an effective PLA-modified filler. However, due to the characteristics of the mixing process and foaming process, its lower expansion rate is difficult to be applied in practical production compared to the high expansion rate of PLA foams reported by other researchers.

Ren et al. [70] also carried out the study of cellulose nanofibers (CNF) to improve the rheological and crystallization behavior of PLA. An overwhelming problem to be overcome in modifying PLA by nanofibers is the dispersion of nanofibers in PLA. Of interest, the authors first used acetylated modified CNF. After the modification, CNF changed from highly hydrophilic to hydrophobic, which overcame the problem that the fibers tend to agglomerate in PLA with polarity. Then, the PLA/CNF composites were prepared by solution blending to further ensure the nanofiber dispersion problem. Finally, it was formed under a high-pressure vessel with CO_2_ as a physical foaming agent, saturation pressure of 1.5 MPa, and foaming temperature of 100 °C to 140 °C. Compared with CNF, the introduction of modified CNF resulted in better foaming performance of PLA. At low pressure (1.5 MPa) and foaming temperature of 120 °C, the foam containing 2 wt% of modified CNF had an expansion ratio of up to 20 times. Oluwabunmi et al. [52] found that PLA compounded with microcellulose fiber (MCF) reduced the T_g_ of PLA. The reason for this is the plasticizing effect exhibited by MCF on PLA, which, at high pressure, increases the migration ability of PLA molecular chains. The PLA foam with 1.5 wt% MCF content showed a significant increase in open cell and porosity and a decrease in thermal conductivity compared with pure PLA, which demonstrated that the PLA/MCF composite foams produced could be applied to zero energy consumption buildings. Also, the authors confirmed the ability of MCF to accelerate the degradation of PLA.

Zhang et al. [94] prepared the composite foam using cotton fiber as the reinforcing agent and poly(ethylene glycol) (PEG) as the lubricant. The foam was prepared by microporous injection foaming process using nitrogen as the foaming agent, and it was found that both impact and stretchability properties were reduced after foaming. Compared with the pure PLA foamed material, the crystallization ability of PLA was slightly improved with the low content of cotton-fiber, and the impact properties of the foam were slightly improved. The addition of high content of cotton fibers, however, reduced the reduced impact and tensile properties. The reason is that the high content of cotton fibers leads to an increase in crystallization ability, which is not conducive to foaming. In contrast, poor fiber dispersion agglomeration causes a decrease in mechanical properties. Therefore, the research to improve the foaming behavior and mechanical properties of PLA/fiber foams focuses on how to solve the problem of the dispersion of nanofibers in PLA. Poor dispersion will directly affect the foam morphology, and it is difficult to improve the pore morphology even by the subsequent foaming process and parameters. Zhang et al. [69] reported a new process, the Pickering emulsion templating method, for preparing poly(lactic acid)/cellulose nanocrystal (CNC) composite masterbatches, Figure 7. The master batches made by the emulsion method exhibited good dispersion behavior with less agglomeration compared with the direct twin-screw blending method. Additionally, the master batches made by the emulsion method exhibited more uniformly distributed cells, smaller cell sizes (~50 µm), and high cell density after foaming (Figure 8). In addition, in this work, the authors also studied the foaming processing window of PLA/CNC foams. They analyzed the effect of foaming time on the bubble morphology to provide thoughts for industrial production applications.

From some of the above studies, it can be seen that nanofiber-modified PLA is an effective way to obtain PLA foams with good cell morphology and mechanical properties. However, the problem of dispersion of fiber filler in PLA should be concerned. The current research shows that the main idea to solve the dispersibility problem of nanofibers in PLA is to reduce the hydrophilic ability of fibers so that they can be better dispersed in PLA with hydrophobicity. The specific idea is first to modify the nanofibers to convert hydrophilic to hydrophobic [70]. It has also been reported that polymer grafting and silylation can effectively improve the aggregation of nanoparticles [96,97]. The second idea is to solve the problem of easy agglomeration by a new mixing process [69], and the above-mentioned emulsion method provides us with a fresh idea.

The modification of PLA by carbon-based nano-fillers is also a hot topic of current research. Nano-fillers can improve the foaming ability of PLA foams on the one hand. On the other hand, they can also give the foam material electrical and thermal conductivity to broaden the application range. Li et al. [95] investigated the rheological behavior and foaming ability of PLA/CNTS composite foams using carbon nanotubes (CNTS). The addition of CNTS significantly improved the rheological behavior of PLA (3 orders of magnitude increase in storage modulus and 2 orders of magnitude increase in complex viscosity) with a volume expansion rate of 49.6 times. This is mainly attributed to the addition of carbon nanotubes, which generate three network structures in the composites, resulting in a significant increase in melt strength (Figure 9). At the same time, carbon nanotubes act as heterogeneous nucleation and improve the crystallization ability. Another paper [62] reported the use of carbon nanotubes (CNTS) and graphene (GP) as reinforcing agents to modify PLA to prepare composite foams synergistically. PLA/CNTS/GP composites were prepared by solvent blending method, and foaming experiments were carried out at a saturation pressure of 20 MPa, at the saturation pressure. It was found that 1D carbon nanotubes and 2D graphene exhibited better rheological behavior, crystallization behavior, and cell morphology than single carbon-based fillers, indicating that the two carbon-based fillers played a synergistic effect, while the electrical conductivity improved to 1.28 x S/m. It is worth mentioning that in general, 2D fillers exhibit better interfacial bonding with PLA than 1D fillers due to their excellent bonding [62], better ability to improve the rheological behavior of PLA, and better foam morphology. Previously, PLA/carbon nanotube (CNTS)/carbon black (CB) composite foams were also prepared using supercritical CO_2_. CNTS and CB did not have a synergistic effect on PLA’s rheological and crystallization behavior.

Guo et al. [61] found that halloysite nanotubes (HNTs) can effectively improve the rheological behavior of PLA and broaden the foaming window of PLA. The compressive modulus of the foams was improved about three times compared with that of pure PLA foams. Interestingly, the authors performed annealing treatment after foaming further to improve the compressive properties of PLA/HNTs foams and found that after annealing, the compressive modulus of pure PLA foams more than doubled, and the PLA/HTNs composite foams did not improve much.

Li et al. [55] investigated the enhancement of PLA microcellular foams with acyl hydrazide nucleating agent (HNA) as a nucleating agent. The addition of HNA was found to form a self-assembled network with a highly crystalline structure, resulting in dense microporous foams. Meanwhile, the authors found that adding this nucleating agent significantly promoted the dissolution of CO_2_ in PLA during the foaming process. By preparing PLA/talc foam, Lin et al. [98] determined that the size of the foam was related to the CO_2_ solubility, and the lower the solubility, the larger the cell volume, which is consistent with the idea of the plasticizing effect of CO_2_. Bruetting et al. [13] also established that the addition of talc effectively improves the problem of the uneven distribution of cells after PLA chain expansion.

### 3.3. Blend Modification

Blending PLA with other polymers is a simple and effective method of modification. By blending with other polymers, the rheological behavior of PLA can be improved, and the deficiencies of some properties of PLA can be corrected. Table 4 shows the research from the last three years, based on scCO_2_ foaming technology, on blends modified with polylactic acid foam. Li et al. [56] prepared open-cell foam materials with good selective oil absorption (7.9–21.9 g/g for grease adsorption) by melt-blending PBS with PLA using supercritical CO_2_ foaming technology. Open-cell foam refers to the appearance of penetrating bubble holes at the interface of the polymer during the foaming growth process. The opening mechanism of open-cell foams [99] prepared based on supercritical fluid foaming has been studied [100,101,102] previously and is not presented in this review. Generally speaking, microporous foams with high porosity are suitable for preparing applications with functional scenarios such as adsorption, filtration, sound absorption, and separation. Foams with high open porosity often require low melt strength, but low melt strength significantly negatively impacts the structure and mechanical properties of PLA foams. Therefore, how to balance the porosity and mechanical properties is a direction to study the open cell foam. Li et al. investigated the cell structure and adsorption of PLA/PBS foam and obtained high porosity (about 97.7%), high expansion ratio (43.6), and high repeatability in selecting oil-absorbing open cell materials. Ren et al. [64] blended polyethylene glycol (PEG) with PLA and used supercritical foaming technology to prepare porous scaffolds. It was found that PEG has good biocompatibility with PLA, and the addition of PEG and appropriate foaming process parameters could reduce the melt strength and improve the hydrophilicity of PLA. The porosity of the porous scaffold obtained with 40 wt% PEG at a foaming pressure of 20 MPa and a foaming temperature of 115 °C reached 95.2% because the hydrostatic pressure, rather than the CO_2_ plasticizing effect [103], dominates the foam morphology when the PLA foaming pressure is greater than 20 MPa. Meanwhile, the authors further conducted biocompatible experiments on PLA/PEG open-cell scaffolds. They found that these scaffolds have ultra-high hydrophilicity and good cytocompatibility, which are expected to be widely used in biological tissues.

To obtain porous materials with a good shape effect, Chai et al. [68] obtained specimens by co-blending different contents of poly(methyl methacrylate) (PMMA) and PLA through a twin-screw extruder. They prepared porous structural materials by a two-step method using CO_2_ as the foaming agent. The results showed that the two were well-compatible after blending, and the matrix produced microcrystals with shape memory capacity. At the same time, the foam with reduced cell size (15 µm), high expansion ratio (up to 28 times), and good compressive resilience was achieved.

### 3.4. Others

In the last three years, in addition to the above studies on modified PLA foams, many researchers have improved the pore morphology and foam properties by inducing PLA crystallization through environmental parameters (pressure, temperature) and the crystallization-induced transformation of PLA spherical crystal structure into a three-dimensional composite structure, as shown in Table 5.

Li et al. [106] used pre-crystallization and supercritical CO_2_ batch foaming techniques to prepare linear PLA open-cell materials. The adsorption-diffusion behavior of CO_2_ in PLA was investigated to probe the foam behavior of PLA at different foaming temperatures and pressures. The results showed that the pre-crystallized PLA could produce a lamellar or non-lamellar structure. With increasing CO_2_ saturation temperature (temperature variation in the range of 125 °C to 143 °C), the PLA microstructure transitioned from non-lamellar to lamellar to non-lamellar. In contrast, the cellular structures of the surface and core of the specimens differed. At lower saturation temperatures (below 125 °C), the degree of melting is low, the crystallinity is high, the material is rigid with low foaming capacity, and the temperature increases (increases to 134 °C), the melt strength increases, the foaming capacity increases, and the temperature reaches a certain level, the melt strength weakens. The cells are more likely to aggregate and collapse. As for the blameless PLA foams, both the skin and core layers showed highly interconnected open pore structures in the range of 24.1 to 31.0 MPa. The adsorption amounts of 2–4 g/g for organic solvents and 0 g/g for water provide a solvent-free preparation method for biodegradable oil-absorbing foam materials.

Wang et al. [58] used batch foaming and analyzed the dissolution behavior of CO_2_ in PLA, and prepared high expansion multiples with good selective oil absorption by saturated foaming after pre-melting PLA at different foaming temperatures and foaming. Open-cell foam with a high expansion multiplier and good selective oil absorption was prepared by the pre-melting and saturated foaming of PLA at different foaming temperatures. PLA foamed after pre-melting to obtain foams with high expansion ratios. Foaming under high-pressure CO_2_ enables PLA to crystallize at lower temperatures. As described earlier, the foaming process, with an increase in temperature, will decrease the melt strength of PLA will decrease, and the cell walls will be thinned by the biaxial stretching of the cell walls, with rupture openings or even collapse.

To investigate the effect of CO_2_ dissolution in PLA during foaming on the crystallization behavior of PLA, Li et al. [107] investigated the crystallization behavior of PLA under high-pressure CO_2_. They found that the effect of CO_2_ on the cellular structure of PLA was mainly due to the decrease in the density of the PLA molecular chains and the weakening of chain interactions by CO_2_, which increased the mobility of chains and local PLA crystallization nucleation, affecting the crystal structure and crystallization kinetics. The presence of local high-pressure CO_2_ may restrict the growth of crystals to the amorphous region, resulting in growth toward the z-axis (direction perpendicular to the crystal surface) or the formation of lamellae parallel to the original nucleus. Yu et al. [51] used density flooding theory (DFT) to investigate the foaming mechanism of the PLA foaming process. The results showed that the density of PLA pores was related to the crystallinity and increasing the crystallinity appropriately could increase the cell density.

Chen et al. [53] used pressure-induced foaming to prepare PLA foams and investigated the effects of different saturation pressures (8–24 MPa) on the cell structure and mechanical properties. The results showed that high saturation pressure (>20 MPa), quickly formed the open pore structure, while the mechanical properties showed a rising enhancement and then weakening with the increase of saturation pressure. Yan et al. [105] reported a study on the preparation of foams with different contents of asymmetric poly(d-lactide) (PDLA) added to pure poly(l-lactide) (PLLA). The introduction of PDLA caused the system to produce a stereo-complex high melting point SC crystals, as non-homogeneous nucleation sites, can enhance the strength of PLA melt and resist cell rupture at high foaming temperatures to some extent. The authors further investigated the properties of PLLA/PDLA foams under the action of SC crystals and found a 1.5-fold increase in the compressive modulus of the composite foam at 3 wt% PDLA. Standau et al. [12] investigated the foaming process of PLA with different levels of D-lactide monomer. The analysis showed that PLA with lower D content needs to be foamed at higher temperatures to obtain suitable foams.

### 3.5. Conclusions

Pure PLA has a typical spherical crystal structure with a narrow foaming window because with the increase in foaming temperature, the molecular chain migration rate is large, and the crystallization shows the result of increasing first and then decreasing, and the nucleation rate is reduced. Utilizing modifications such as chain expansion, filler enhancement, blending, and induced crystallization can promote the nucleation rate, facilitate the formation of cells, and widen the foaming window. In addition, it can also regulate the melt strength and stabilize the cell structure to improve the foam performance. Chain expansion or cross-linking is an effective way of modification, but the prepared foam is generally not fully degradable. Nanoscale filler is an excellent nucleating agent, and the method of filler modification can prepare high-density micro- and nanoscale cells, but how to better disperse the nanoscale filler in PLA matrix to uniformly distribute good mechanical properties with available foam is still a challenge. The current article is a feasible idea to prepare PLA/particle filler composites by solution or emulsion method. For fiber filler, acetylation treatment is also an improvement idea. The blending of PLA with polymers is simple and suitable for commercialization. However, the selection of suitable blends and the improvement of the compatibility of the blends are difficulties related to the blend modification, and the poor compatibility directly affects the final product’s performance. It is a frontier research direction to improve PLA’s rheological behavior and foaming behavior by self-induced, pressure-induced, temperature-induced, and other procedures without introducing nucleating agents or other fillers under the compatibility and particle agglomeration between PLA composites. The appropriate foaming temperature, foaming pressure, and pressure relief rate for directly improving the crystallization behavior of pure PLA as well as establishing the solubility of CO_2_ in PLA and the function of the diffusion mechanism remains for future research; however, it has been reported in the literature [53,64] that when the saturation pressure reaches 20 MPa or more, the plasticizing effect of CO_2_ does not play a dominant role in the growth of bubble pores. More models are needed to determine the effect of saturation pressure.

## 4. Applications of PLA Composite Foams

In recent years, research on CO_2_-assisted foamed PLA has focused on regulating PLA rheological and crystallization behaviors, and some researchers have also focused on the application properties. Table 6 summarizes the performance studies of PLA foams prepared based on scCO_2_ foaming in the last three years. PLA microporous foams have excellent application prospects. Firstly, PLA has good biocompatibility and excellent mechanical and thermal properties, making its application in multiple fields possible. Secondly, the microporous foam is lightweight, and the open-cell microporous foam has good absorption and penetration properties, promising for sound absorption and noise reduction, filtration and adsorption, and biological tissue engineering. The closed-cell microporous foam is also widely used in cushioning, packaging, construction, and aerospace.

Li’s team used PLA, PLA, and PLA/PBS as systems to prepare microporous foams of these systems by pre-crystallization combined with CO_2_ pressure induction [106], pre-melting combined with CO_2_ pressure induction [57], and co-mingled intermittent foaming methods [56], respectively. The process conditions for forming open-cell foams were investigated and foaming under high pressure. All obtained foams with a high open-cell ratio (all >90%) and good selective adsorption ability for different organic solvents and greases. The superior adsorption capacity of open-cell foams is attributed to the highly interconnected network structure and the micro- and nano-sized bubble pore morphology. This complex of tiny network channels can store organic solvents very well. Of interest is the excellent adsorption capacity and adsorption repeatability of PLA/PBS open-cell foams for organic solvents >20 times, providing a feasible solution for applying apparent oil adsorption. Geopolymer/PLA composite foam was prepared by Tommasini et al. [110] and had good adhesion to heavy metal ions (Cu and Zn).

Open-cell microporous materials obtained by green scCO_2_ foaming have gained wide attention for biological tissue engineering applications. High open-cell PLA/PEG porous scaffolds were prepared by Ren et al. [64]. The possibility of porous scaffolds for tissue engineering applications was explored by examining the survival of cells on the scaffolds. The porous scaffolds were found to be very biocompatible with cells in vitro. Cells could adhere and grow well on the porous material. Zhukov et al. [109] reported a modified PLA porous material with a piezoelectric effect, and this iron column polar body could be applied to biological tissue engineering. PLA microporous scaffolds have great promise for tissue engineering applications [111]. PLA’s good biocompatibility and biodegradability suit the essential biological criteria for tissue engineering applications. Meanwhile, batch foaming technology and high-pressure injection molding technology can achieve high porosity and interconnectivity to meet cell attachment, proliferation, and growth requirements. Optimizing the foaming process to achieve a high void fraction and uniform scaffolds is crucial for future PLA microporous materials in tissue engineering.

Closed-cell microporous foam prepared by supercritical foaming has good mechanical properties and has a promising future in architectural decoration and aerospace. Xiang et al. [65] prepared PLA with an oriented crystal structure using pressure-induced flow (PIF) and then prepared PLA foam with intermittent foaming. In the PIF process, the crystals are compressed in one direction to form a selectively oriented micro fibrillated structure, Figure 10. The crystal structure is transformed from a spherical crystal to a fiber structure with orientation. Micro/nanoscale cells are obtained after foaming (nanoscale cells are formed within the microfibers, and microscale cells are formed between the lamellae), as shown in Figure 11. The authors conducted impact and tensile property tests on the foams obtained at different foaming temperatures. It was found that the size of both micro- and nanocells decreased, and their number increased, when the temperature was increased. Foaming at temperature 40 resulted in foams with superior impact properties and tensile properties (Figure 12). The good performance is believed to be the result of the action of the bimodal micro/nanocellular structure formed after foaming by the fiber structure with orientation obtained by the PIF process. Interestingly, the PIF process can broaden the foaming window of PLA, and the micro and nano bimodal cell structure can impart better sound and vibration damping properties to the material. From the foaming process point of view, the PIF process provides an effective way to prepare porous materials with excellent properties. In one of the authors’ articles [59], it was also demonstrated that the foam with micro-nano-bimodal structure has excellent impact performance.

In addition, PLA microporous materials can also be applied in the field of battery shielding. Xu et al. [112] prepared PLA/poly(chlorinated biphenyl)/carbon nanotube CNTs porous composites, and the CNT was modified to achieve de-electromagnetic shielding effectiveness of 41 dB.

## 5. Conclusions and Prospects

PLA foams are lightweight, green, and biodegradable, and they can be applied in a wide range of uses such as sound absorption and heat insulation, cushioning and packaging, oil absorption and filtration, and tissue engineering. The development of PLA foam has scientific and practical significance to broadening the field of commercial plastic foams, which is currently limited to petroleum-based polymer foams mainly and is an effective, environmentally friendly foam application.

However, the study of PLA foams still has some challenges due to the rheological and thermal behavior of PLA and more directly its low melt strength and slow crystallization rate on the foaming results. The complexity of dissolution, adsorption, and diffusion of CO_2_ in PLA based on supercritical CO_2_-assisted foaming technology for the preparation of PLA foams also causes uncertainties in the foaming behavior. The common idea of current research in this field is to study the relationship between the behavior of CO_2_ dissolution in PLA and the foaming parameters (saturation pressure, foaming temperature, etc.) and the microstructure and properties of the foam, and this relationship has not yet established a complete theory to explain. At present, the cell morphology and structure of PLA are overcome by the way of modification ideas, and PLA modification work is also the most important direction of PLA foam research in the last three years. The introduction of nucleating agents can overcome the low melt strength and low crystallization rate of PLA to some extent, but the introduction of nucleating agents can also create challenges in understanding the results of specific PLA foams, while also considering the compatibility and dispersion issues of the modification process. The preparation of PLA by induced crystallization without the addition of nucleating agents is also a research direction. Optimizing the foaming process is another important way to improve the cell morphology and properties of PLA and its composite foams. The improvement of batch foaming and the establishment of multistage foaming schemes are increasingly investigated. For example, in situ fibrillation or pressure-induced flow is performed before foaming to orientate the crystals and improve the material’s mechanical properties. Of course, the perfect foaming process still needs further research. Giving more functionality for commercial applications while satisfying good foam morphology is worthy of attention. Finally, most of the research on PLA foams has been conducted through intermittent foaming, and their industrial commercialization is a challenge. In considering the efficiency of foam production, there is a need to optimize the foaming process and enrich PLA foaming technology. The high degree of freedom and lightweight properties of 3D printing foaming technology is expected to have great potential for future development. Industrial production of PLA foam is anticipated to be better solved in the future.

## Figures and Tables

**Figure 1 polymers-14-04320-f001:**
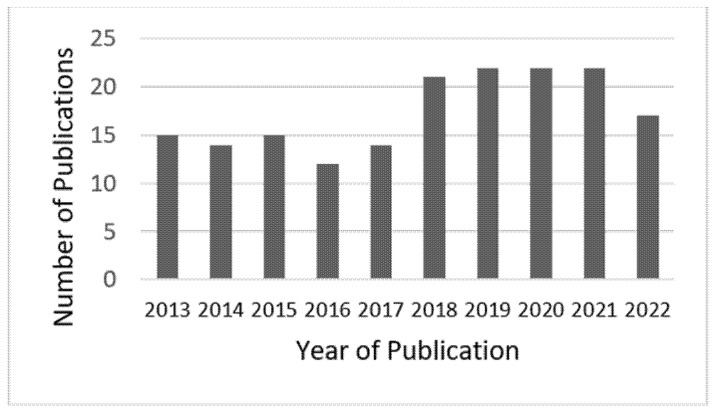
Publication volume of supercritical CO_2_ assisted polylactic acid foaming from 2012 to 2022. Source: Web of Science.

**Figure 2 polymers-14-04320-f002:**
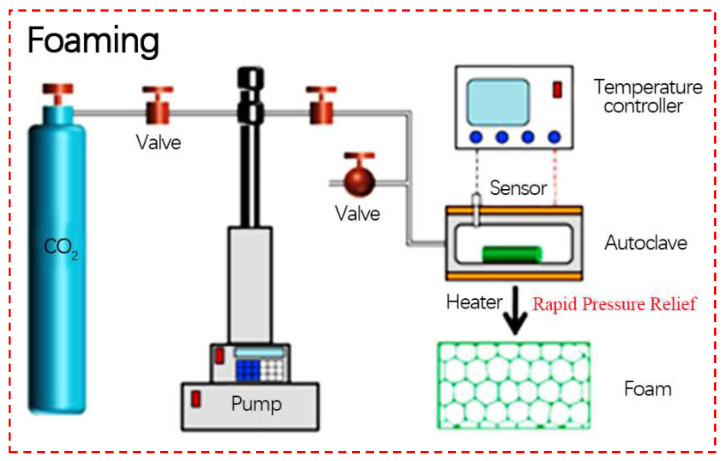
Pressure-induced foaming process (reproduced with permission from Reference [56], Copyright 2022 Elsevier).

**Figure 3 polymers-14-04320-f003:**
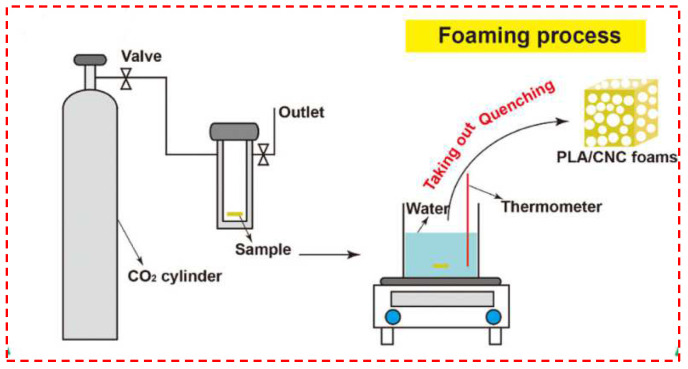
Principle of the temperature-induced batch foaming process (reproduced with permission from Reference [69], Copyright 2021 American Chemical Society).

**Figure 4 polymers-14-04320-f004:**
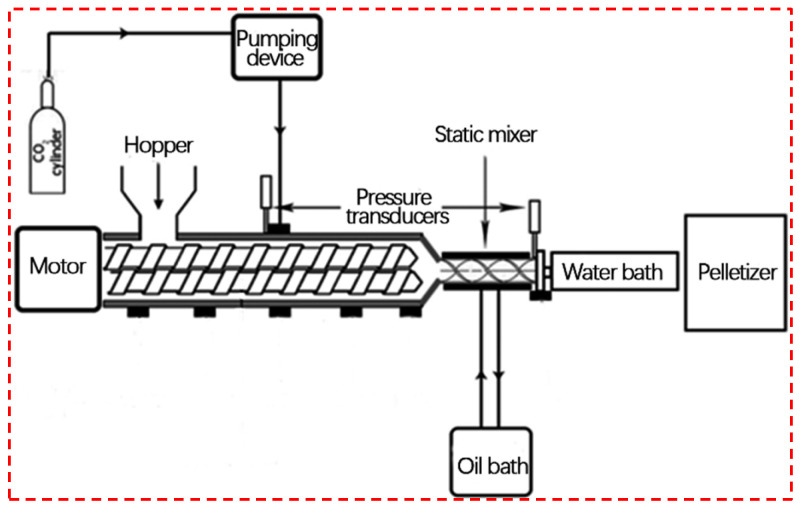
Principle of extrusion foaming process (reproduced with permission from Reference [73], Copyright 2021 Elsevier).

**Figure 5 polymers-14-04320-f005:**
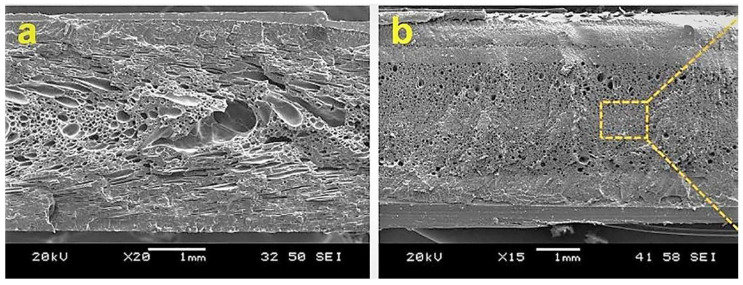
Scanning electron microscopy (SEM) image of foamed samples: (**a**) pure PLA foam prepared by regular micropore injection molding, (**b**) pure PLA foam High-pressure microwell injection molding (reproduced with permission from Reference [79], Copyright 2022 Elsevier).

**Figure 6 polymers-14-04320-f006:**
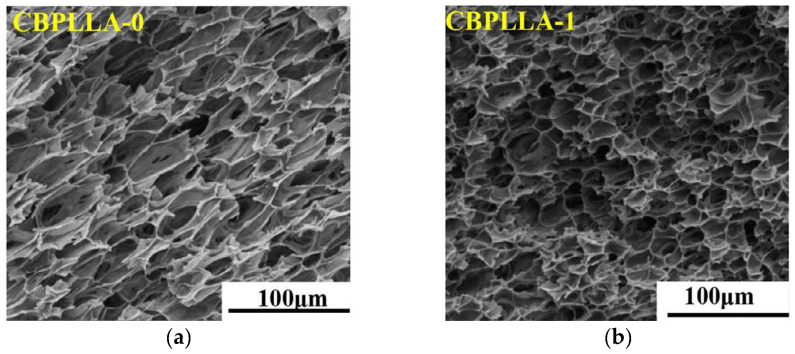
SEM images of branched PLLA and branched PLLA/TMC foam at 130 °C: (**a**) pure PLLA; (**b**) 1 wt% TMC-300; (**c**) 3 wt% TMC-300 (reproduced with permission from Reference [91], Copyright 2022 Elsevier).

**Figure 7 polymers-14-04320-f007:**
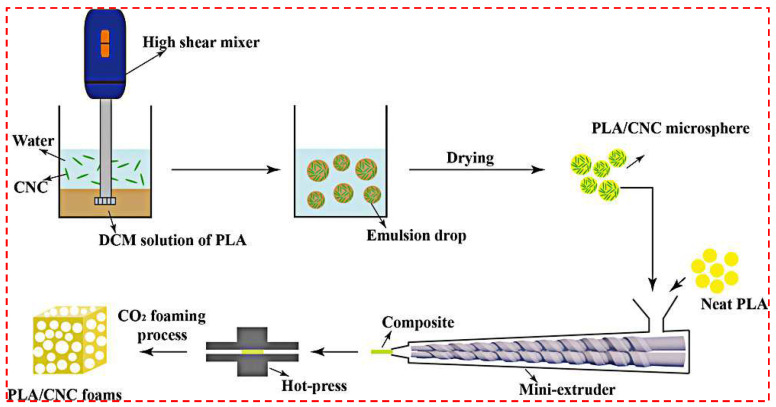
Principle of PLA/CNC composite films prepared by Pickering emulsion method (reproduced with permission from Reference [69], Copyright 2021 American Chemical Society).

**Figure 8 polymers-14-04320-f008:**
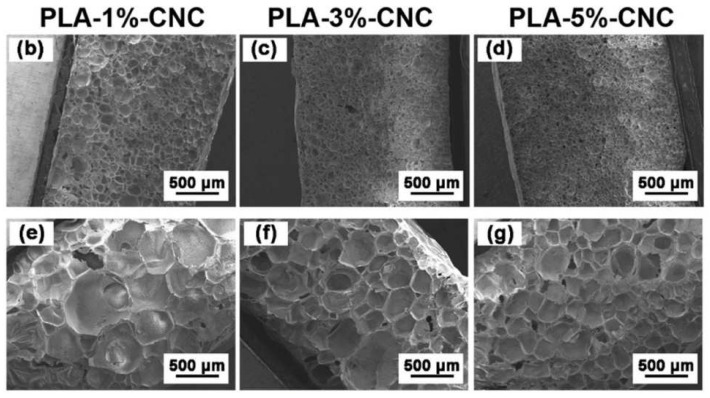
SEM images of cross-sectioned PLA/CNC composite foams with different CNC content (up to 5 wt%): (**b**–**d**) prepared by the Pickering emulsion masterbatch route; (**e**–**g**) prepared by direct melt blending (reproduced with permission from Reference [69], Copyright 2021 American Chemical Society).

**Figure 9 polymers-14-04320-f009:**
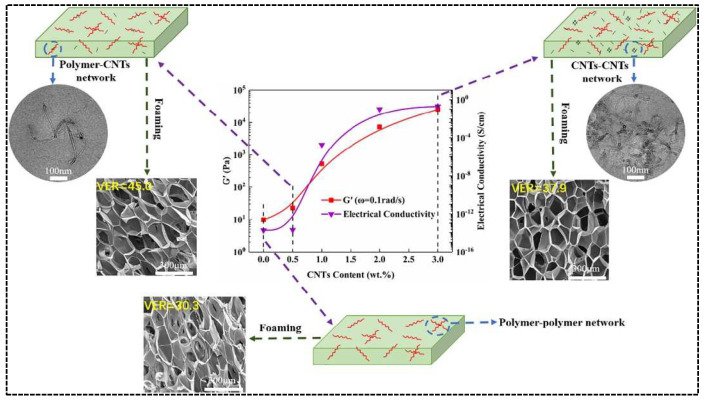
Schematic diagram of three possible network structures of PLA/CNTS foams (reproduced with permission from Reference [95], Copyright 2020 Elsevier).

**Figure 10 polymers-14-04320-f010:**
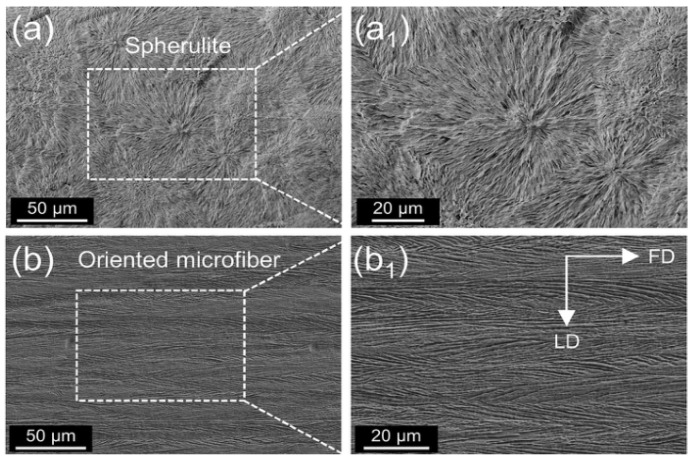
Typical PLA crystal structure: (**a**) crystal obtained by cold crystallization at 120 °C; (**b**) directional crystal structure formed by pressure induced flow; (**a_1_**,**b_1_**) are local enlarged images of the corresponding crystal. FD: flow direction, LD: load direction (reproduced with permission from Reference [65], Copyright 2022 Elsevier).

**Figure 11 polymers-14-04320-f011:**
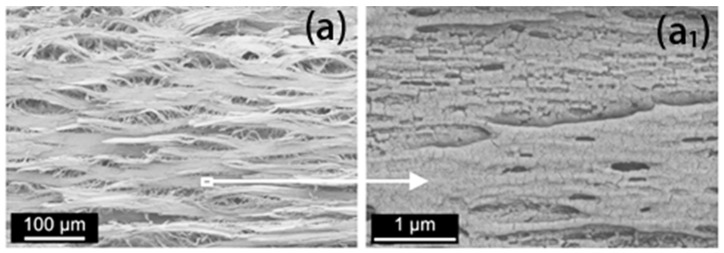
Image of polylactic acid foaming micro/nano bimodal cell structure after pressure induced flow. (**a**) SEM images of nano-sized cells obtained at foaming temperature of 60 °C; (**a_1_**) SEM images of nano-sized cells from the local observation of (**a**) (reproduced with permission from reference [65], Copyright 2022 Elsevier).

**Figure 12 polymers-14-04320-f012:**
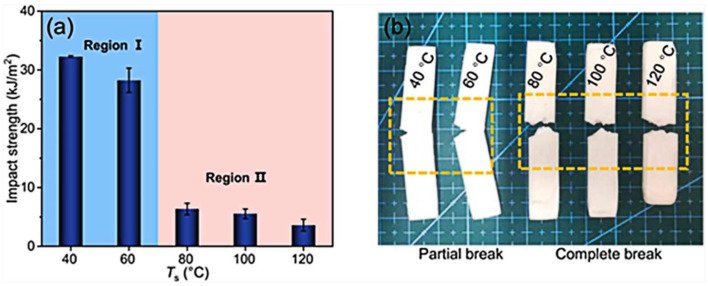
(**a**) Impact strength and (**b**) typical impact fracture photos of PLA foam samples with pressure-induced flow at different foaming temperatures (reproduced with permission from reference [65], Copyright 2022 Elsevier).

**Table 1 polymers-14-04320-t001:** Research work on the preparation of PLA and its composite foam materials by batch foaming method in the past 3 years. (PLA is from NatureWorks, USA. when not otherwise stated.).

Research System	PLA Brand	Method	Volume Expansion Ratio	Cell Size	Cell Density	Reference
PLA	2003D	Pressure-induced foaming	--	~40 µm	~10^11^ cell/cm^3^	[51]
PLA/MCF	4060D		~5	24 µm	~3 × 10^7^ cell/cm^3^	[52]
PLA	4032D		4.1	6.4 µm	~10^9^ cell/cm^3^	[53]
PLA/CE	2003D		~10	40 nm	10^15^ cells/cm^3^	[54]
PLA/HNA	3053D		~6 (reduce)	~6.59µm	~10^9^ cell/cm^3^	[55]
PLA/PBS	4032D		~58	158 µm	--	[56]
PLA	4032D		~58	~50 µm	2 × 10^7^ cell/cm^3^	[57]
PLA	4032D		~39.2	~150 µm	2 × 10^8^ cell/cm^3^	[58]
PLA	2003D		--	40 µm/180 nm	--	[59]
PLA/SiO_2_ aerogel/CE	3052D		~55 (reduce)	120 µm	0.024 g/cm^3^	[60]
PLA/HNTs	4032D		~19	~150 µm		[61]
PLA/CNTs/CB	4032D		~3	~13 µm	10^8^ cell/cm^3^	[62]
PLLA	PLA by Total Corbion		24.8	40 µm	10^8^ cell/cm^3^	[63]
PLA/PEG	2003D		9.1	~55 µm	~10^7^ cell/cm^3^	[64]
PLA	4032D		1.4	30 µm/70 nm	2 × 10^6^ cell/cm^3^/ 2 × 10^11^ cell/cm^3^	[65]
PBST/PLA	4032D		17	7.1 µm	3.5 × 10^10^ cell/cm^3^	[66]
PLA/PDLA/PBS	4032D	Temperature-induced foaming	--	0.66 µm	--	[67]
PLA/PMMA	4032D		~28	15 µm	~10^7^ cell/cm^3^	[68]
PLA/CNC	4060D		--	~50 µm	5 × 10^6^ cell/cm^3^	[69]
PLA/CNF	TP-4000 by UNITIKA Ltd.		20.4	<40 µm	>10^8^ cell/cm^3^	[70]
CA/PLA	4032D		--	0.6 µm	~10^12^ cell/cm^3^	[71]

**Table 2 polymers-14-04320-t002:** Research on chemical modification of polylactic acid-based foams from 2020 to 2022.

Research System	PLA Brand	Expansion Ratio	Cell Size	Cell Density	Reference
PLA/CE (Epoxy-based styrene-acrylic acid oligomers)	2003D	~10	40 nm	10^15^ cells/cm^3^	[54]
PLA/CE (a random copolymer of ethylene and glycidyl methacrylate)	2003D	~42(↑)	~155 µm	~10^16^ cells/cm^3^	[90]
PLA/CE (Acrylic acid oligomer)	2003D	11	~10 µm	10^10^ cell/cm^3^	[91]
PLA/EPDM/CL (1,3,5-Triallyl-1,3,5-triazine-2,4,6 (1H,3H,5H)-trione)	4032D	~28	40 µm	10^9^ cell/cm^3^	[81]
PLA/SiO_2_ aerogel/CE (MDI)	4043D	~55	120 µm	0.024 g/cm^3^	[60]

In the table, “CE” refers to chain extender and “CL” refers to cross-linking agent.

**Table 4 polymers-14-04320-t004:** Study on blending modification of PLA foam in recent three years.

Research System	PLA Brand	Expansion Ratio	Cell Size	Cell Density	Reference
PLA/PBS	4032D	58	158 µm	--	[56]
PLA/PMMA	4032D	~28(↑)	15 µm	~10^7^ cell/cm^3^	[68]
PLA/EPDM/CL	4032D	~28	40 µm	10^9^ cell/cm^3^	[81]
PLA/PEG	2003D	9.1	~55 µm	~10^7^ cell/cm^3^	[64]
TPU/PLA	4032D	7.1	~30 µm	--	[104]

In the table, “CL” refers to cross-linking agent.

**Table 5 polymers-14-04320-t005:** Study on PLA foam by induced crystallization in the last three years.

Research System	PLA Brand	Expansion Ratio	Cell Size	Cell Density	Reference
PLLA/PDLA	4032D	~48	250 µm	6 × 10^4^ cell/cm^3^	[105]
PLA	4032D	4.1	6.4 µm	~10^9^ cell/cm^3^	[53]
PLA	2003D	--	~40 µm	~10^11^ cell/cm^3^	[51]
PLA	4032D	--	40 µm/180 nm	--	[65]
PLA	4032D	~39.2	~150 µm	2 × 10^8^ cell/cm^3^	[58]
PLA	4032D	~58	~50 µm	2 × 10^7^ cell/cm^3^	[57]
PLA	4032D	1.4	30 µm/ 70 nm	2 × 10^6^ cell/cm^3^/ 2 × 10^11^ cell/cm^3^	[59]
PLLA	PLA Lx175	24.8	40 µm	10^8^ cell/cm^3^	[63]

**Table 6 polymers-14-04320-t006:** Study on the properties of PLA foam prepared by scCO_2_ foaming in the last three years.

Functionality	Application	Research System	Performance	Reference
Mechanical property	Aerospace /Buffer packaging	PLA/HNA	High compressive strength	[55]
		PLLA/PDLA	Good compressibility	[106]
		PLA/HNTs	Good compressibility	[61]
		PLA/PMMA	Shape memory	[68]
		PLA/CNTs/CB	Good pressure resistance	[62]
		TPU/PLA	Shape memory	[104]
		PLA/EDPM	Super toughness	[81]
		PLA	Good toughness	[65]
		PLA	Good impact toughness	[59]
Heat preservation/insulation	Packaging /Construction	PLLA/PDLA	Thermal conductivity: 31.2 mW/(m·k)	[105]
		PLA/PET	Thermal conductivity: 26.8 mW/(m·K)	[79]
		PLA/HNTs	Thermal conductivity: 34.29 mW/(m·K)	[61]
		PLA/EPDM/CL	Thermal conductivity: 26.33 mW/(m·K)	[81]
		PLA/SiO_2_ aerogel	Thermal conductivity: 26.28 mW/(m·K)	[60]
		PLA	Thermal conductivity: 31.8 mW/(m·K)	[57]
Adsorption/Filtration	Filtration	PLA	Adsorption capacity of CCl_4_ 15 g/g	[58]
		PLA	Oil absorbency: 10.9–31.2 g/g	[57]
		PLA/PBS PLA	Oil absorbency: 7.9–21.9 g/g Adsorption capacity of CCl_4_ 5 g/g	[56] [106]
Biocompatibility	Biological tissue engineering	PLA/PEG	Sertoli cell attachment	[64]
		PLA/Ketoprofen	Drug release	[108]
		PLA/PCL	Hydrophilic; support Cell growth	[72]
		Modified PLA	Piezoelectric effect	[109]
Conductive	Container/Sensor	PLA/CNTs/CB	Electrical conductivity: 1.28 × 10^−1^ S/m	[62]

## Data Availability

Not applicable.

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
