# Peer review of "Progress in the Preparation, Properties, and Applications of PLA and Its Composite Microporous Materials by Supercritical CO2: A Review from 2020 to 2022"

_polymers, 2022, doi:10.3390/polym14204320_

Round 1

Reviewer 1 Report

Comment: The review presents the progress obtained during the last 3 years on PLA foaming using supercritical carbon dioxide. This paper is interesting and will be useful to the scientific community. It is well-organized.  English is poor and difficult the understand. Some Figures have poor quality (8 and 9, for example). Moreover, some points to be corrected are listed below.

Lines 43-45: The sentence is incomplete.

The end of the introduction should be written with sentences instead of a list of topics.

Before entering into the details of the different foaming technology, the authors may explain the important characteristics of the foam (cell density, interconnectivity, cell size, define what is volume expansion ratio,…)

Line 68 versus Line 126: The authors said contradictory statements. They explain that the formation and growth of bubbles are triggered by “pressure relief” whereas they said that “a sudden increase in […] pressure” is necessary in Line 68.

In table 1, the column “Method” should be filled, or the authors must explain why it is not.

Figure 2 is more related to the batch foaming apparatus than to a description of the process. The legend should be modified.

Line 184: Water may not be the best medium for this technique applied to PLA since PLA suffers hydrolysis. The authors should comment on it.

Line 194: Please correct the sentence “blending PLA and PLA”.

Line 251: The term “distance” is not common in the injection process. Please, be more precise.

Line 278 and paragraph 421-430: What do open-mold mean? Where/when is UV applied? The term “in-situ fibrillation” is unclear in that case: is EPDM in a fiber shape?

Line 334: substitute “alloy” by “blend”.

Line 368 and 383: remove the space between poly and (lactic acid)

Line 392-393: this sentence is confusing.

Table 3: All the abbreviations must be explained (MCF, CNF, CNC, CB, HNA…) in a footnote.

Generally, when authors talk about mechanical properties, they should be more precise (Elastic modulus? Stress at break?...). A similar comment can be made regarding the generic “rheological properties” mentioned.

Line 646: What is called “PLA vesicles”?

Line 651: the authors should explain to what “z-axis” they refer to.

Line 662: what is poly(D-propyleneglycolate)? Is it the same as PDLA? Why not use a more common name for PDLA? Same for PLLA.

Author Response

Dear Reviewer:

    Thank you for the reviewer's comments concerning our manuscript entitled " Progress in preparation and properties of PLA microporous materials by sCO2 foaming technology: A Review" (Manuscript ID: polymers-1923851). Those comments are all valuable and very helpful for revising and improving our paper. We have studied comments carefully and have made correction which we hope meet with approval. The corrections are marked in red in the manuscript (See the attachment).The main corrections in the paper and the responds to the reviewer's comments are as flowing:

    1.The review's comment: English is poor and difficult the understand.

The authors' answer: Thank you for your suggestions on the language of our manuscript, your comments are thoughtful. We have rechecked the language of the manuscript based on the original text. The sentences and grammar were corrected and marked in red in the manuscript.  We tried our best to improve the manuscript and made some changes in the manuscript.

     2. The review's comment: Some Figures have poor quality (8 and 9, for example).

The authors' answer: We have made changes based on the reviewers' comments. For Figure 8 and Figure 9 regarding poor image quality, we have re-selected high-resolution images for replacement, and we especially thank you for your good comments.

    3. The review's comment: Lines 43-45: The sentence is incomplete.

The authors' answer: We thank the reviewers for their careful review of the manuscript, and we have made this sentence more complete to reduce the ambiguity caused by this statement. The ester group on the molecular chain of PLA makes it have good degradability." was placed earlier in the paragraph and added appropriately. See lines 46 to 50.

    4. The review's comment: The end of the introduction should be written with sentences instead of a list of topics.

The authors' answer: We thank the reviewer for their suggestion on the format of the manuscript, which we have benefited from. We have revised the subheading at the end of the introduction to present it in paragraph form. See lines 127 to 132.

    5.The review's comment: Before entering into the details of the different foaming technology, the authors may explain the important characteristics of the foam (cell density, interconnectivity, cell size, define what is volume expansion ratio,…)

 The authors' answer: Thanks to the reviewer' suggestions on the content of the manuscript, which were very useful to us. For the definition of parameters important for describing the foam structure: cell size, cell density, and volume expansion ratio we have added in the second part foaming technique. See lines 150 to 157.

    6. The review's comment: Line 68 versus Line 126: The authors said contradictory statements. They explain that the formation and growth of bubbles are triggered by “pressure relief” whereas they said that “a sudden increase in […] pressure” is necessary in Line 68.

The authors' answer: We thank the reviewer for the correction on the principle of supercritical foaming. After careful comparison, we have made a more precise representation of cell nucleation and cell growth during foaming. See lines 85 to 89, lines 165 to 173.

    7. The review's comment: In table 1, the column “Method” should be filled, or the authors must explain why it is not.

The authors' answer: We would like to thank the reviewer's additional suggestions on the column "Method" in Table 1 of the manuscript. In Table 1, we have added a special column for "Method" in order to provide a summary of which intermittent foaming (rapid pressure relief (pressure-induced foaming) and rapid temperature rise (temperature-induced foaming) have been used by researchers in the last three years. In Table 1 we summarize and represent these two methods in two parts according to each of them. This is expressed in the column "Methods".

    8. The review's comment: Figure 2 is more related to the batch foaming apparatus than to a description of the process. The legend should be modified.

The authors' answer: We thank the reviewer's valuable suggestions regarding the appropriateness of the content of the images cited in the manuscript. We humbly accept the reviewer's comments, and perhaps Figure 2 is more of a description of the foaming equipment. However, after further consideration, we believe that this picture combined with the textual description is able to convey the intent of the pressure-induced intermittent foaming process. The picture depicts the equipment for intermittent foaming and a process of pressure-induced foaming. In addition, we have added an illustration of "Rapid pressure release" to the diagram.

     9. The review's comment: Line 184: Water may not be the best medium for this technique applied to PLA since PLA suffers hydrolysis. The authors should comment on it.

The authors' answer: Thanks to the reviewer's valuable suggestions. This issue was rarely mentioned in relevant articles this year. PLA is susceptible to hydrolysis, and hydrolysis may cause a change in the molecular weight of PLA, resulting in a decrease in performance, even in a very short period of time. We briefly comment on the treatment of water bath foaming based on the reviewer's comments. See lines 232 to 235.

  10.The review's comment: Line 194: Please correct the sentence “blending PLA and PLA”.

The authors' answer: Thank you for your careful review. This has been revised in a timely manner and the phrase "by blending PLA and PLA and" has been removed. See line 244.

    11. The review's comment: The term “distance” is not common in the injection process. Please, be more precise.

The authors' answer: Thank you for your suggestion, "distance" does have a vague definition. We have replaced it with "mold opening distance". See line 314. It should be noted that the "mold opening distance" is one of the most important factors affecting the bubble morphology in the high-pressure microcellular injection foaming process. This is explained in the manuscript.

    12. The review's comment: Line 278 and paragraph 421-430: What do open-mold mean? Where/when is UV applied? The term “in-situ fibrillation” is unclear in that case: is EPDM in a fiber shape?

The authors' answer: Thank you for your question about the above words. We did not explain it very well in our presentation. Now, based on your query, we have made the following changes to the above.

(a). open-mold is a type of injection foam, also known as high-pressure microcellular injection foam, whose definition is described in line 307 to 309.

(b). UV applications are also described in particular, see lines 342 to 347.

(c). The expression "in situ nanofibrillation" appears several times in the application article, where it refers to EPDM nanofibrillation by a melt-spinning process.

(d). EPDM is a fibrous structure formed by melt spinning and the raw material is not a fibrous structure. See lines 342 to 347.

    13. The review's comment: Line 334: substitute “alloy” by “blend”.

The authors' answer: Thank you for your suggestion. We have replaced the word "allloy" with "blend". See line 403.

    14. The review's comment: Line 368 and 383: remove the space between poly and (lactic acid)

The authors' answer: We thank you for pointing out the problem with the expression of the professional names, and we have checked the format of the names of the polymers in this manuscript. We have also revised it accordingly. line 437, line452.

     15. The review's comment: Line 392-393: this sentence is confusing. Line 646: What is called “PLA vesicles”?

The authors' answer: The two issues identified above have been revised accordingly in a timely manner. See line 475 to 478, line735. Thank you for pointing them out.

    16.The review's comment: Table 3: All the abbreviations must be explained (MCF, CNF, CNC, CB, HNA…) in a footnote.

The authors' answer: Thank you for your comments on the manuscript regarding abbreviations. We have included a note summarizing all the abbreviations in this paper and have included them in the introduction section. See lines 109 to 110.

    17. The review's comment: Generally, when authors talk about mechanical properties, they should be more precise (Elastic modulus? Stress at break?...). A similar comment can be made regarding the generic “rheological properties” mentioned.

The authors' answer: We thank the reviewer's valuable comments. Expressing the specific mechanical properties already rheological characteristics can make the article clear within the content. We have now expressed the mechanical properties and rheological characteristics in a concrete way. For example, line98,247,249,327,569.

    18. The review's comment: Line 651: the authors should explain to what “z-axis” they refer to. Line 662: what is poly(D-propyleneglycolate)? Is it the same as PDLA? Why not use a more common name for PDLA? Same for PLLA.

The authors' answer: Thanks to the reviewer for asking the question. z-axis direction is perpendicular to the direction of the crystal surface. See line 729.

Also the whole process of PLLA and PDLA has been revised and explained. See line 741 for the basic way of writing them.

We appreciate for Reviewer's warm work earnestly and hope that the correction will meet with approval. Once again thank you very much for your comments and suggestions.

Yours sincerely

Kangming Peng

Reviewer 2 Report

Generally speaking, the paper is intented to be a review but it is not. I see on the figure titles that the paper review is limited to recent years ?, 2015-2022 ?). It is a more summary of some works carried out on PLA foams, or more precisely on PLA blended with other compounds or PLA composites (am I right ?).

The titre says "microporous", while the paper is not focussing on microporosity. Nothing is said on the combination of PLA and microporosity (which has to be defined), which can be interesting for a review or a paper.

In the title, Technology should be at plural "technologiES" (batch, extrusion, etc).

Bio degradability is put forward several times, while PLA is not properly speaking self degradable, but it needs specific industrial conditions (e.g T).

The manuscript has indeed a significant number of references (108) but they are often not relevant or out of scope or not used at the right place. For example, ref 1,2,3,4 are not representative of what the manuscript's text says..., ref 1 and 2 not with PE foams; ref3 not with PS.  Ref 5-7 are out of place here , not repersentative of classical PMMA foams. Ref  8 is out of scope. so on and so on; all references need to be revised or checked. A key word research gives a recent review not analysed here : Foaming of PLA Composites by Supercritical Fluid-Assisted Processes: A Review , Jennifer Andrea Villamil Jiménez et al. Molecules, 2020,

Manuscript's inconsistencies are numerous, examples of :

line 48 ('melt strength low) VS line 51 ('melt strength is too high'), I do not understang these lines, should I read high or low ?.

line 54 : dense bubble pores is what ? => pore density ?

line 68: increase ? => decrease 

...

...

For me, the text stands descriptive, rather general. Nothing is really analysed for the relation between process/technology <> porosity type (e.g micro) <> foam density. This would be worth a review.

Focus the manuscript on one aspect instead of stating a review.

I would say that either the mansucript needs quite major revisions (or even should be rejected).

Author Response

Dear Reviewer:

    Thank you for the reviewer's comments concerning our manuscript entitled " Progress in preparation and properties of PLA microporous materials by sCO2 foaming technology: A Review" (Manuscript ID: polymers-1923851). Those comments are all valuable and very helpful for revising and improving our paper. We have studied comments carefully and have made correction which we hope meet with approval. The corrections are marked in red in the manuscript (See the attachment). The main corrections in the paper and the responds to the reviewer's comments are as flowing:

    â‘ The review's comment: Generally speaking, the paper is intented to be a review but it is not. I see on the figure titles that the paper review is limited to recent years ?, 2015-2022 ?). It is a more summary of some works carried out on PLA foams, or more precisely on PLA blended with other compounds or PLA composites (am I right ?).

The authors' answer: We are very grateful to the reviewer's suggestions on the framework of the manuscript. Our intent in writing this review is primarily to review the last 3 years of progress on CO2-assisted PLA foams, rather than to provide a systematic overview of the progress of PLA foams. Since there are previous reviews of past PLA foams, the idea of our work is expressed in the abstract and at the end of the introduction, see lines 102 to 132. In addition, we briefly answer the reviewer's question about whether the system studied in the manuscript is a PLA foam or a PLA and its composite foam. It is our understanding that PLA foam, as it should be, may not only refer to pure PLA foam, but it can more broadly include modified PLA foam. The summary of our work on PLA foam is done on PLA and its composites. In the presentation we simply use "PLA foams", which may be ambiguous, so we are open to the reviewers' comments. The title of the manuscript has been more precisely qualified regarding the object of the work and the scope of the work.

    â‘¡The review's comment: The titre says "microporous", while the paper is not focussing on microporosity. Nothing is said on the combination of PLA and microporosity (which has to be defined), which can be interesting for a review or a paper.

The authors' answer: We appreciate the reviewers' suggestions on the manuscript regarding "microporous". This was very helpful to us in improving the manuscript. For supercritical CO2 foaming technology, it is actually a microporous foaming technology, which has been reported in previous literature and books. It is because of the connection between supercritical CO2 and microporousness that we use the term "PLA microporous material" in the title. Based on the reviewer's comments, we revised and added to the manuscript, adding the relationship between supercritical foaming technology and microporosity in lines 77 to 81. We also added the definition of microporous. We are very grateful to the reviewers for their suggestions on the content of our manuscript, which will help us to improve the manuscript.

    â‘¢The review's comment: In the title, Technology should be at plural "technologies" (batch, extrusion, etc).

The authors' answer: Thanks to the reviewer's help with the manuscript regarding grammar. The word "technology" in the title should indeed be expressed in the plural form, and we have corrected it in time. We have also corrected the word "technology" in the abstract and text. See lines 29, 35, 131, 134.

    â‘£The review's comment: Bio degradability is put forward several times, while PLA is not properly speaking self-degradable, but it needs specific industrial conditions (e.g T).

The authors' answer: As suggested by the reviewer, we have expressed the biodegradability of PLA several times in the manuscript, however without a more complete expression of biodegradability. Thanks to the reviewer's comments, we have added these in the manuscript. See lines 45 to 50.

    ⑤The review's comment: The manuscript has indeed a significant number of references (108) but they are often not relevant or out of scope or not used at the right place. For example, ref 1,2,3,4 are not representative of what the manuscript's text says..., ref 1 and 2 not with PE foams; ref3 not with PS.  Ref 5-7 are out of place here , not repersentative of classical PMMA foams. Ref  8 is out of scope. so on and so on; all references need to be revised or checked.

The authors' answer: Many thanks to the reviewer's suggestions on the cited texts. As the Reviewer suggested that we checked the cited literature again and found that some of it may be outside the working topic or beyond the topic. We reworked the references [1-8]. The literature of the entire manuscript was also reexamined and added or removed.

    â‘¥The review's comment: A key word research gives a recent review not analysed here : Foaming of PLA Composites by Supercritical Fluid-Assisted Processes: A Review , Jennifer Andrea Villamil Jiménez et al. Molecules, 2020,

The authors' answer: Thank the reviewer’s suggestions on citations. We have cited this review in the revised manuscript, see Ref. [43], line 103.

    ⑦The review's comment: line 48 ('melt strength low) VS line 51 ('melt strength is too high'), I do not understang these lines, should I read high or low ?. line 54 : dense bubble pores is what ? => pore density ? line 68: increase ? => decrease

The authors' answer: In view of these valuable suggestions, we have revised the manuscript.

 First, regarding "line 48 ('melt strength low) vs. line 51 ('melt strength is too high')", PLA is characterized by low melt strength. The low melt strength limits the foaming of PLA, so the melt strength has to be regulated to increase the PLA melt strength. However, too high melt strength is not always beneficial, which may affect the expansion rate of the foamed material. Possibly due to the semantic confusion brought about by the presentation problem, we have revised this section, see lines 60 to 65.

Second, about "line 54 : dense bubble pores is what ? => pore density ?" This is a low-level error in our expression, where we want to express the meaning of "cell density". Thank you for pointing this out, and we have also rechecked the whole article to standardize the relevant names.

Third, about "line 68: increase ? => decrease", we were going to say "a sudden rise in temperature or a sudden drop in pressure". Thank you for pointing out that we have rephrased the foaming principle accurately, see lines 85 to 89.

    â‘§The review's comment: For me, the text stands descriptive, rather general. Nothing is really analysed for the relation between process/technology <> porosity type (e.g micro) <> foam density. This would be worth a review.Focus the manuscript on one aspect instead of stating a review.

The authors' answer: Thank you for your comments on the framework and logic of our manuscript. For us your suggestions are very constructive. The connection between porosity type and process is probably not an aspect that we will focus on in this manuscript. This may be an interesting point of discussion for our future work. The framework of this manuscript is relatively simple, a review of the last three years of PLA foam research work in terms of foaming technology, modification methods and applications, and a commentary on this work. The review is given in the course of the review as well as in the summary. The review may not be extremely insightful, but it also gives suggestions and ideas for current PLA foam research. Your questions and suggestions are very helpful for my future work, and I thank you again for reviewing the manuscript.

    We appreciate for Reviewer's warm work earnestly and hope that the correction will meet with approval. Once again thank you very much for your comments and suggestions.

Yours sincerely

Kangming Peng

Reviewer 3 Report

This manuscript reviews the recent publications on the polylactic acid foams with supercritical carbon dioxide blowing agent. The overall idea is good and the manuscript has been relatively well written. An extensive literature survey has been conducted and there is a systematically layout in the manuscript. However, the manuscript needs some major revisions before final decision.  

1. It is recommended not to use the abbreviations in the title.

2. It is recommended to include the “applications” in the title as “Progress in preparation, properties and applications of ...”

3. Line 68 “Then, a sudden increase in temperature or pressure leads to supersaturation of the gas, triggering cell nucleation and growth.” This sentence has serious problem. This is not a step in the foaming process. Supersaturation of the blowing agent is conducted before the process (in batch process) or through the process (in extrusion and injection processes). Nucleation and growth of the bubbles are the main steps of the polymeric foaming process. Nucleation is triggered by sudden increase in temperature or by sudden drop in pressure. Growth is triggered by penetration of the gas into the bubbles. However, the mentioned sentence covers none of these facts. Please revise it carefully.

4. It is recommended to include the most recent review papers in the field of polymeric foams in the references; my recommendations are https://doi.org/10.1016/j.cej.2021.132662, https://doi.org/10.1016/j.jmmm.2021.168038, https://doi.org/10.3390/ma13184060, https://doi.org/10.1177%2F0021955X20959301, https://doi.org/10.1080/00914037.2022.2066669, https://doi.org/10.1016/j.polymer.2022.124681.

5. Although there are many lumped references in the manuscript, some of them such as [45-50] are not acceptable.

6. Line 242 “Microporous injection molding foam technology (MIM)” should be changed to “Microporous injection molding (MIM) foam technology”.

7. One of the main weaknesses of the manuscript is Table 1. Providing this information without any discussions or justifications is not reasonable.

8. If possible, it is recommended to provide the chemical modifier in separate column in Table 2.

Author Response

Dear Reviewer:

     Thank you for the reviewer's comments concerning our manuscript entitled " Progress in preparation and properties of PLA microporous materials by sCO2 foaming technology: A Review" (Manuscript ID: polymers-1923851). Those comments are all valuable and very helpful for revising and improving our paper. We have studied comments carefully and have made correction which we hope meet with approval. Meanwhile, the English language and style were modified. The corrections are marked in red in the manuscript (See the attachment). The main corrections in the paper and the responds to the reviewer's comments are as flowing:

    â‘ The review's comment: It is recommended not to use the abbreviations in the title.

The authors' answer: Thank you for your suggestions on our manuscript, your comments were very useful to us. We have removed the abbreviated words from the title of the manuscript based on your suggestions.

    â‘¡The review's comment: It is recommended to include the “applications” in the title as “Progress in preparation, properties and applications of ...”

The authors' answer: Thank you for your thoughtful comments on our manuscript title suggestions. We feel that the addition of "Applications" is necessary in the context of the manuscript. Based on your suggestion, we checked the original title and then redefined the current manuscript title and marked it in red in the manuscript. We tried our best to improve the manuscript and made some changes in the manuscript.

    â‘¢The review's comment: Line 68 “Then, a sudden increase in temperature or pressure leads to supersaturation of the gas, triggering cell nucleation and growth.” This sentence has serious problem. This is not a step in the foaming process. Supersaturation of the blowing agent is conducted before the process (in batch process) or through the process (in extrusion and injection processes). Nucleation and growth of the bubbles are the main steps of the polymeric foaming process. Nucleation is triggered by sudden increase in temperature or by sudden drop in pressure. Growth is triggered by penetration of the gas into the bubbles. However, the mentioned sentence covers none of these facts. Please revise it carefully.

The authors' answer: We thank the reviewer for the correction on the principle of supercritical foaming. After careful comparison, we have made a more precise representation of cell nucleation and cell growth during foaming. The presentation of the principle elsewhere in the manuscript was also checked. See lines 85 to 89 and lines 165 to 173.

    â‘£The review's comment: It is recommended to include the most recent review papers in the field of polymeric foams in the references; my recommendations are https://doi.org/10.1016/j.cej.2021.132662, https://doi.org/10.1016/j.jmmm.2021.168038, https://doi.org/10.3390/ma13184060,

https://doi.org/10.1177%2F0021955X20959301, https://doi.org/10.1080/00914037.2022.2066669,  https://doi.org/10.1016/j.polymer.2022.124681.

The authors' answer: We thank the reviewer's suggestions regarding literature citation in the manuscript. The reviewer's recommendation of several recent reviews on PLA foams is, in our opinion, very relevant to the improvement of our manuscript. As we understand it, four of these reviews are very relevant to our topic. We finally cited them.

doi:10.1016/j.cej.2021.132662 See reference [48] line 158;

doi.org/10.3390/ma13184060 see reference [50] line 161;

doi.org/10.1177%2F0021955X20959301 , see reference [28] line 80;

doi.org/10.1016/j.polymer.2022.124681, see reference [111] line 814;

    ⑤The review's comment: Although there are many lumped references in the manuscript, some of them such as [45-50] are not acceptable.

The authors' answer: We thank the reviewer's suggestions regarding literature citation in the manuscript. The purpose of our application at that time was to illustrate the article on N2 foam. In conjunction with the reviewer's comments we felt that it was a bit inconsistent with the topic of our article and therefore removed references that were confusing. See line 148. Also, based on the reviewers' comments, we re-examined the suitability of the selected literature and finalized the citation of the existing literature.

    â‘¥The review's comment: Line 242 “Microporous injection molding foam technology (MIM)” should be changed to “Microporous injection molding (MIM) foam technology”.

The authors' answer: We thank the reviewer's meticulous review. We have revised this issue, see line 295.

    ⑦The review's comment: One of the main weaknesses of the manuscript is Table 1. Providing this information without any discussions or justifications is not reasonable.

The authors' answer: We greatly appreciate the reviewers' suggestions for Table 1 of the manuscript. We then supplemented Table 1 in this section. The purpose was to provide a summary of the intermittent foaming (rapid pressure relief (pressure-induced foaming) and rapid warming (temperature-induced foaming) used by researchers in the preparation of PLA foams over the past three years. We first based this table on the introduction of intermittent foaming techniques and then supplemented it. In response to the reviewer's comment "without any discussions or justifications is not reasonable.", we felt that it was justified, so we briefly discussed (or explained) the table in the article, see line 178 to 180. Of course a more specific discussion is necessary, and this is the focus of this section of the manuscript, see 2.1 batch foaming technology.

    â‘§The review's comment: If possible, it is recommended to provide the chemical modifier in separate column in Table 2

The authors' answer: We are very grateful to the reviewer's suggestions on Table 2 of the manuscript. The description of chemical modifiers is necessary, and we are glad that the reviewer gave me these suggestions, and we have made the full statement about chemical modifiers in Table 2 of the revised manuscript. See line 435 to 436.

    We appreciate for Reviewer's warm work earnestly and hope that the correction will meet with approval. Once again thank you very much for your comments and suggestions.

Yours sincerely

Kangming Peng

Round 2

Reviewer 1 Report

The paper is now suitable for publication.

Reviewer 2 Report

I see that corrections were made; many of them answer the comments and questions of the first report. I agree with the corrections suggested. The paper is more defined and have limits in "time and topics". The manuscript still contains some misunderstandings and lacks to be accepted as a review paper. To my mind, the paper can be accepted after these corrections.

1- Still, for me, the word "microporous" needs to be retrieved from the title. I insist that most of the time, the PLA foams described are not micro porous,  but most of the time they are macro porous. Therefore, just use the word "porous" or "foams", and do not use wherever in the text "micro porous materials".

2- The reference below is now cited. I had a look again and it seems to me rather complete. It is from 2020, indeed it is right in the scope and the period of the manuscript. So the progress or the new look of your manuscript from that date can be given in 3 or 4 sentences. Your manuscript  claims to be a review , so it has to be an overall analysis ; noveties need to be brought, most of all if you analyse the bibliography only from 2020.

Foaming of PLA Composites by Supercritical Fluid-Assisted Processes: A Review ,
Jennifer Andrea Villamil Jiménez; Nicolas Le Moigne; Jean-Charles Bénézet; Martial Sauceau; Romain Sescousse, Molecules, Vol 25, Iss 3408, p 3408 (2020) DOI: 10.3390/molecules25153408

Reviewer 3 Report

The authors have carefully addressed my comments and modified the manuscript accordingly. Based on the Reviewer’s opinion, the revised manuscript deserves the publication.